# Post-traumatic growth in mental health recovery: qualitative study of narratives

Mike Slade,[1] Stefan Rennick-Egglestone,[1] Laura Blackie,[2]
Joy Llewellyn-Beardsley,[1] Donna Franklin,[3] Ada Hui,[1] Graham Thornicroft,[4]
Rose McGranahan,[5] Kristian Pollock,[6] Stefan Priebe,[5] Amy Ramsay,[4] David Roe,[7]
Emilia Deakin[1]

For numbered affiliations see end of article.

**Correspondence to**
Professor Mike Slade;
m.slade@nottingham.ac.uk

## ABSTRACT

**Objectives** Post-traumatic growth, defined as positive psychological change experienced as a result of the struggle with challenging life circumstances, is under-researched in people with mental health problems. The aim of this study was to develop a conceptual framework for post-traumatic growth in the context of recovery for people with psychosis and other severe mental health problems.

**Design** Qualitative thematic analysis of cross-sectional semi-structured interviews about personal experiences of mental health recovery.

**Setting** England.

**Participants** Participants were adults aged over 18 and: (1) living with psychosis and not using mental health services (n=21); (2) using mental health services and from black and minority ethnic communities (n=21); (3) underserved, operationalised as lesbian, gay, bisexual and transgender community or complex needs or rural community (n=19); or (4) employed in peer roles using their lived experience with others (n=16). The 77 participants comprised 42 (55%) female and 44 (57%) white British.

**Results** Components of post-traumatic growth were present in 64 (83%) of recovery narratives. Six superordinate categories were identified, consistent with a view that post-traumatic growth involves learning about oneself (self-discovery) leading to a new sense of who one is (sense of self) and appreciation of life (life perspective). Observable positively valued changes comprise a greater focus on self-management (well-being) and more importance being attached to relationships (relationships) and spiritual or religious engagement (spirituality). Categories are non-ordered and individuals may start from any point in this process.

**Conclusions** Post-traumatic growth is often part of mental health recovery. Changes are compatible with research about growth following trauma, but with more emphasis on self-discovery, integration of illness-related experiences and active self-management of well-being. Trauma-related growth may be a preferable term for participants who identify as having experienced trauma. Trauma-informed mental healthcare could use the six identified categories as a basis for new approaches to supporting recovery.

**Trial registration number** ISRCTN11152837

## INTRODUCTION

Post-traumatic growth is a relatively new area of research. The concept was introduced 20

### Strengths and limitations of this study

► This study reports findings from a substantial qualitative investigation of post-traumatic growth in people with psychosis and other severe mental health problems.
► The purposive sampling involved under-researched groups, including people with mental health problems who do not use mental health services or who are underserved by mental health services.
► The involvement of eight analysts from varied backgrounds including several with lived experience of mental ill-health increases the credibility of the data analysis.
► Participants were self-selecting so the views of people who do not associate with the term 'recovery' may be under-represented.
► Participants were not asked specifically about trauma experiences.

years ago[1] and is defined as perceptions of 'positive psychological change experienced as a result of the struggle with highly challenging life circumstances'.[2] Other terms for this phenomenon have also been used, such as benefit-finding,[3] both a coping profile and a coping outcome[4] and positive illusions.[5] This range of terms points to the complexity of the phenomenon, with different theorists emphasising change in identity and narrative (ie, the sense of self),[6] change in eudaemonic well-being (ie, subjective quality of life)[7] and change in social/psychological resources (ie, broadening response repertoires).[8] However, there is broad consensus over five post-traumatic growth domains: improved relations with others, identification of new possibilities for one's life, increased perception of personal strengths, spiritual growth and enhanced appreciation of life.[2]

Post-traumatic growth is now a well-established concept in relation to 'event trauma', that is, experience of a single traumatic event. Post-traumatic growth is widely reported in relation to event trauma and it is

associated with mental health outcomes, such as depression. However, the direction of relationships between post-traumatic growth and depression are not consistent across studies. For example, post-traumatic growth was found to moderate the negative association with quality of life found for both depression and post-traumatic stress in 58 Norwegian survivors interviewed 2 and 6 years after the 2004 Southeast Asia tsunami.[9] Other studies have shown that post-traumatic growth is associated with a higher level of depression, such as a 2-year follow-up study of 316 survivors of the L'Aquila earthquake in Italy.[10] Similarly, higher post-traumatic growth was associated with lower post-traumatic stress disorder and higher depression in 186 Iraqi students with an average of 13 war-related adversity experiences.[11] There is some evidence of a curvilinear association between depression and post-traumatic growth, for example, two studies of assault survivors (n=270 in total) found survivors with low or high post-traumatic growth reported more symptoms of post-traumatic stress (both studies) and depression (one study) than those with intermediate growth levels.[12] Longitudinal research is needed to fully understand how reports of post-traumatic growth interact with depression in the recovery from event trauma.

Post-traumatic growth has relevance to healthcare, for example, liver transplantation[13] and stroke.[14] There are several reasons why post-traumatic growth may also be relevant to psychosis and other severe mental health problems.[15] First, abuse incidence is high. In particular, childhood adversity is strongly associated with increased risk of psychosis. A meta-analysis of 18 case–control studies (n=2048 psychosis, n=1856 controls), 10 prospective studies (n=41 803) and 8 population-based cross-sectional studies (n=35 546) found significant associations between adversity and psychosis across all designs (OR 2.78, 95% CI 2.34 to 3.31).[16] Second, comorbid depression, which is implicated in post-traumatic growth, is common. For example, around 50% of people with a diagnosis of schizophrenia also experience depression.[17] Third, both the experience of psychosis itself and the consequent experiences of societal discrimination and retraumatisation caused by mental health system responses[18 19] may generate trauma. Finally, some people with personal experience of psychosis report post-traumatic growth.[20 21]

There is strong evidence that positive changes can be experienced after first-episode psychosis. A systematic review,[22] published initially as a scoping review,[23] identified 40 studies involving 715 participants investigating the experience of positive change after first episode psychosis. The review identified three levels of positive change. The individual-level change theme was developed by combining three sub-themes: insight and clarity (eg, reassessing one's life, realising who one's friends are, less emphasis on materialism and societal expectations); personality, outlook and skills (eg, more authenticity, better able to handle stress, learning time management); and health, lifestyle and interests (eg, simpler life, new possibilities, better sleep habits). The interpersonal-level change theme was developed by combining two sub-themes: relationships with family and friends (improvements in communication, spending more time together, letting go of unhealthy relationships) and place or role in society (desiring to give back, challenging stigma in society). The religious or spiritual-level theme examples were praying more, increased engagement in religious institutions and positive religious experiences.

There is only limited evidence about the frequency and types of post-traumatic growth in people living long-term with psychosis and other severe mental illness experiences.[24] A quantitative study of 121 people with severe mental illness using community mental health rehabilitation centres in Israel found high levels of trauma,[25] and that meaning-making and coping self-efficacy mediated post-traumatic growth experiences.[26] Three small (n=7,[27] n=7,[28] n=10[29]) qualitative studies using interpretative phenomenological analysis of semi-structured interviews all identified themes of personal growth.

The concept of recovery in mental health has come to mean living as well as possible, whether or not symptoms are present. Recovery thus differs from the traditional clinical priorities of symptom remission and functional restoration,[30] and a systematic review identified five recovery processes: Connectedness, Hope, Identity, Meaning and Empowerment (CHIME Framework).[31] Current evidence indicates that growth is integral to recovery, and involves both restoration of existing aspects of identity and construction of new aspects.[32]

The extent to which (a) the five growth processes identified from event trauma research and (b) the more preliminary early psychosis-specific restorative and constructive identity processes are characterising the same changes is unknown. Integration of these two sources of evidence is needed, as is investigation of the experiences of a broader range of people with long-term psychosis, including under-researched groups. The aim of this study is to develop a conceptual framework for post-traumatic growth in the context of recovery for people with psychosis and other severe mental health problems.

## METHODS

This research was undertaken as part of the National Institute for Health Research (NIHR) Narrative Experiences Online (NEON) study between March and August 2018.

### Participants

A purposive sample of under-researched populations took part. Inclusion criteria common to all groups were: aged over 18, willing to discuss experiences, able to give informed consent and fluent in English.

Group A (outside the system) comprised people having had self-identified experiences of psychosis of sufficient frequency or duration that they identify as someone with experience of psychosis, and not using services. Psychosis experience prevalence estimates in the general population range from 7%[33] to 13%,[34] yet lifetime rates

of psychosis, determined through contact with services, range from 0.2% (narrowly defined criteria) to 0.7% (broadly defined).[35] The experience of the many individuals who have psychosis without using services may therefore illuminate growth processes. Additional inclusion criteria for group A were: self-identified lifetime experience of psychosis; no use of secondary mental health services over the previous 5 years.

Group B comprised people who identified as being from black and minority ethnic (BAME) populations. Ethnic minority groups often have problematic relationships with services,[36] and research about recovery in these populations identifies a strong emphasis on the post-traumatic growth concepts of connectedness[31] and spirituality.[37] Additional inclusion criteria for group B were: currently using mental health services; black, Asian and minority ethnic community member.

Group C (underserved) comprised people who were not well engaged with mental health services. This was operationalised for three under-served groups: people from lesbian, gay, bisexual or transgender (LGBT+) communities[38]; from rural communities[39]; or with multiple complex health and social care needs.[40] Additional inclusion criteria for group C were: experience of mental health problems in previous 10 years; no or mainly unsuccessful interactions with formal mental health services; member of LGBT+ communities or living in an electoral district area with less than 10 000 population or experience of at least two of homelessness, substance misuse issues or offending.

Group D (peer) comprised people with experience of working in statutory or voluntary roles for which lived experience is a requirement, for example, peer support workers, trainers or researchers. Addition inclusion criteria for group D were: working in statutory or voluntary roles for which lived experience is an essential requirement; use their lived experience as a tool for engagement with other mental health service users.

## Setting

Participants were recruited across England, with groups A (outside the system) and B (BAME) primarily from London and groups C (underserved) and D (peer) primarily from the Midlands. Guidelines for recruitment are shown in online supplementary 1. Group A were recruited through primary care services support groups, Hearing Voices Network and online advertising. Group B were recruited through community groups, a Recovery College and psychosis-specific secondary mental health services. Group C were recruited through community networks, voluntary sector organisations and secondary care mental health services. Group D were recruited through community groups and secondary care mental health services. Recruitment for all groups used snowball sampling.

## Procedures

A preliminary coding framework (online supplementary 2) was developed in advance of interviews, by collating existing post-traumatic growth dimensions identified in trauma populations[41] and in previous studies of post-traumatic growth in psychosis.[22 23 28] Duplicates were removed, dimension names were made consistent and then thematic grouping of the dimensions was used to develop the preliminary coding framework which was intended to establish the link between existing research and participant narratives, and comprised the code name, definition and examples drawn from the source references. An 'other' category was added to allow the emergence of new themes.

Each participant took part in a 40–90 min interview conducted in a health service or community venue. Interviews using a narrative approach were conducted by four researchers from sociology, advocacy, public health and health psychology backgrounds. The topic guide (shown in online supplementary 3) asked the participant to share their mental health and recovery experiences, as if it were a story, with a beginning, a middle and an end, and to include some consideration of what might happen in the future. Participants were remunerated (£20) for their time, and given options to pause or discontinue if they became distressed. Interviews were recorded, and pseudonymised transcripts were made after interviews. After the interview, the researcher wrote field notes which were included in the analysis.

## Analysis

The four coders comprised three interviewers plus one non-interviewer with an interdisciplinary background in sociology and mental health nursing. Thematic analysis was undertaken using NVivo V.11.[42] The study had a predetermined focus on developing a conceptual framework starting from a synthesis of empirical evidence derived from event trauma research as described above which was then developed through analysis of interview data. An approach akin to framework analysis was adopted.[43] Framework analysis involves applying a priori codes and categories to qualitative data to explore specific questions of interest to the research aims, with attention also paid to inductive analysis of data relating to topics which were not anticipated in advance, and to responsive revision of the coding frame as analysis progresses and in the light of regular discussion within the research team.

Coding was initially according to the preliminary coding framework and informed by emergent understandings captured by field notes, but coders remained open to the identification of additional themes in the data. Coding involved identification and allocation of text relating to the coding framework, enabling related text to be grouped and compared, allowing identification of themes occurring within and across sources. Regular discussions between analysts explored how themes of post-traumatic growth were expressed and related to each other, allowing lower order themes to be recognised.[44] Each coder independently coded and compared the same initial transcript. Remaining transcripts were then coded separately (25% per coder).

The coding framework was then iteratively refined in meetings between the four primary coders and a wider group of four other non-interviewer analysts with expertise in healthcare technologies, qualitative research, recovery research and clinical psychology. Several of the interviewers, coders and analysts also had lived experience of mental ill-health and recovery, to enhance the role of lived experience in collection and analysis of data.[45] To enhance trustworthiness, an audit trail was kept, and an interim coding framework is shown in online supplementary 4. The conceptual framework, that is, the final coding framework, was agreed by all coders and analysts.

## RESULTS

The sociodemographic and clinical characteristics of the 77 participants are shown in table 1.

Post-traumatic growth components were coded in 64 (83%) of the 77 transcripts. The conceptual framework for post-traumatic growth in psychosis and other severe mental health problems is summarised in table 2 with

| Table 1 | Clinical and sociodemographic characteristics of participants (n=77) | | | | |
|---|---|---|---|---|---|
| **Characteristic** | **Total** | **Group A (outside the system)** | **Group B (BAME)** | **Group C (under-served)** | **Group D (peer)** |
| n (%) | 77 (100) | 21 (27) | 21 (27) | 19 (25) | 16 (21) |
| Gender n (%) | | | | | |
| Female | 42 (55) | 14 (67) | 11 (53) | 8 (42) | 9 (56) |
| Male | 30 (39) | 6 (29) | 9 (43) | 9 (47) | 6 (38) |
| Other/prefer not to say | 5 (6) | 1 (5) | 1 (5) | 2 (11) | 1 (6) |
| Ethnicity n (%) | | | | | |
| White British | 44 (57) | 12 (57) | 0 (0) | 18 (95) | 14 (88) |
| Black British | 5 (6) | 2 (10) | 3 (14) | 0 (0) | 0 (0) |
| Black African/Caribbean | 4 (5) | 1 (5) | 3 (14) | 0 (0) | 0 (0) |
| White other | 5 (6) | 2 (10) | 1 (5) | 0 (0) | 2 (13) |
| White and black African/Caribbean | 4 (5) | 0 (0) | 4 (19) | 0 (0) | 0 (0) |
| Asian/mixed white Asian | 4 (5) | 0 (0) | 4 (19) | 0 (0) | 0 (0) |
| Other | 5 (6) | 2 (10) | 3 (14) | 0 (0) | 0 (0) |
| Prefer not to say | 6 (8) | 2 (10) | 3 (14) | 1 (5) | 0 (0) |
| Age (years) | | | | | |
| 18–25 | 4 (5) | 0 (0) | 0 (0) | 3 (16) | 1 (6) |
| 25–34 | 16 (21) | 3 (14) | 6 (29) | 4 (21) | 3 (19) |
| 35–44 | 16 (21) | 5 (24) | 4 (19) | 4 (21) | 3 (19) |
| 45–54 | 30 (39) | 8 (38) | 9 (43) | 6 (32) | 7 (43) |
| 55+ | 5 (6) | 4 (19) | 0 (0) | 0 (0) | 1 (6) |
| Prefer not to say | 6 (8) | 1 (5) | 2 (10) | 2 (11) | 1 (6) |
| Sexual orientation | | | | | |
| Heterosexual | 49 (64) | 15 (71) | 14 (67) | 6 (32) | 14 (88) |
| LGBT+ | 18 (23) | 3 (14) | 4 (19) | 9 (47) | 2 (13) |
| Prefer not to say | 10 (13) | 3 (14) | 3 (14) | 4 (21) | 0 (0) |
| Primary diagnosis | | | | | |
| Schizophrenia or other psychosis | 11 (14) | 5 (24) | 4 (19) | 2 (11) | 0 (0) |
| Bipolar disorder/cyclothymia | 16 (21) | 8 (38) | 1 (5) | 3 (16) | 4 (25) |
| Mood disorder, for example, anxiety, depression, dysthymia | 15 (19) | 1 (5) | 4 (19) | 4 (21) | 6 (38) |
| Other, for example, ADHD, personality disorder, substance abuse, autism | 7 (9) | 0 (0) | 2 (10) | 3 (16) | 2 (13) |
| Prefer not to say | 28 (36) | 7 (33) | 10 (48) | 7 (37) | 4 (25) |

ADHD, attention-deficit/hyperactivity disorder; BAME, black and minority ethnic; LGBT, lesbian, gay, bisexual or transgender.

**Table 2** Final conceptual framework for post-traumatic growth in psychosis and other severe mental health conditions

| Type of growth | Definition of the positively perceived change |
| --- | --- |
| 1. Self-discovery | Having a fuller and deeper understanding of oneself. |
| 1.1 Emotional life | Discovering or re-discovering how to access, accept and be mindful of inner emotional life and difficult feelings. |
| 1.2 Self-knowledge | Knowing oneself better, being more authentic and not being as shaped by the expectations of others. |
| 1.3 Self-acceptance | Grieving and letting go of the past, and developing self-compassion. |
| 1.4 Self-responsibility | Taking (back) responsibility for one's own life. |
| 2. Sense of self | Development of a more positive sense of self, including integration and valuing of illness experiences. |
| 2.1 Pride in self | Taking pride in oneself, including personal strengths and achievements. |
| 2.2 Integration of experiences | Illness experiences become an accepted part of one's sense of self. |
| 2.3 Valuing of experiences | Finding positives in the experience of illness. |
| 3. Life perspective | New or renewed appreciation of or gratitude about aspects of life. |
| 3.1 Appreciation of life | Appreciation for life and the importance of hopefulness. |
| 3.2 Appreciation of support | Gratitude for support received from services. |
| 3.3 Meaningful suffering | Gratitude that suffering was meaningful and not in vain. |
| 3.4 Survivor mission | New growth of political consciousness or use of illness experiences to benefit others. |
| 4. Well-being | More active engagement in, and management of, one's own well-being and lifestyle. |
| 4.1 Motivation | Increased determination to stay well, self-manage and not return to a bad situation. |
| 4.2 Being active | More engagement in the arts, music, sport, nature and learning. |
| 5. Relationships | More actively choosing and valuing relationships with others. |
| 5.1 Choosing relationships | Actively choosing relationships to continue, to re-start or to end. |
| 5.2 Valuing relationships | Placing more value on relationships with others. |
| 5.3 Empathy | Enhanced ability to empathise with others. |
| 6 Spirituality | Deeper engagement with spirituality, religious and existential endeavours. |
| 6.1 Spiritual awareness | Increased awareness of the presence of something greater than oneself making a positive contribution by providing meaning. |
| 6.2 Spiritual engagement | New or renewed engagement with spiritual or religious practices, helping with meaning-making and providing comfort and security. |

a complete version including more example coding in online supplementary 5.

## Major themes

### Theme 1: Self-discovery

The self-discovery theme involves a fuller and deeper understanding of oneself. This involves the ability to access, accept and be mindful of difficult feelings.

…what was going on was an internal not an external thing. (B04)

…the key to everything isn't it, accessing your emotions, not running away from them… (C19)

The resulting self-knowledge leads to greater authenticity and being less influenced by the expectations of others.

I feel like I know myself quite well, you know I can heal myself. (A08)

When I discovered that freedom, that I didn't have to join the rat-race…that was quite liberating… (B18)

Self-acceptance can arise from greater self-knowledge, by letting go of past difficulties and developing self-compassion.

It's all about self-accepting, getting to know me and it helped. (A17)

The key word is accepting the situation that I was in, um, and being honest with myself. (C03)

Alongside these processes, participants talked about the importance of taking, or taking back, responsibility for one's own life.

…that hit me like a, er, thunderbolt, knowing that, knowing I, that if I'm not going to help myself, no one else will help me. And that was the beginning, really, of my recovery. (B02)

This was like taking a step back and looking at almost

re-engineering life to take into account self-care, self-preservation and also building myself up… (D08)

## Theme 2: Sense of self

The development of a more positive sense of self involved integration and valuing of illness experiences. A repeated theme was pride in oneself as a person.

I believe in my self-worth these days…I must pat myself on the back. (A15)

I think that's something I'm pretty proud of actually, that I just take people as they are. (B21)

An important part of this pride was integrating experiences of mental ill-health so they become an accepted part of one's sense of self.

You don't choose the issues that you've got but you can, you can make a choice to change. (C04)

I am who I am because of what happened. (D04)

Some participants moved beyond accepting to positively valuing these experiences.

What they were calling symptoms that must be eradicated, were actually part of me and so I looked behind that and said that is where, that is where my creativity comes from. (A19)

I am still me but I am a different me and I am stronger. (D05)

## Theme 3: Life perspective

Participants identified a new or renewed appreciation of, or gratitude about, aspects of life. For some this was a general sense of appreciation.

…I'm becoming one of those ridiculously ever hopeful, ever optimistic people who say there is hope, my life is a life that is about hope… (A08)

…I am alive, I appreciate that I am alive… (C19)

For others there was a new appreciation of support received from support groups, mental health services and workers.

…it [organisation] completely changed my view of life. (A11)

Rehab and coming to <service name> changed my life, it's like <worker> got me on a college course, it has been absolutely wonderful… (C18)

The idea that suffering has been meaningful or worthwhile was expressed.

Wow I see where I am and I go back to then and I just think 'you didn't go through that in vain'. (A17)

…I suppose I am grateful, for want of another way of putting it, that I have lived the life I have, I have had these experiences. (D04)

New connections with political consciousness or a survivor mission to ensure others did not have similar bad experiences were identified by some participants, especially in group B.

…I could identify with more, kind of, politicised… the personal was political and I was beginning to become aware of that on a deeper level. (B09)

And then as I came back out it was just like no, I wanted to help…I've got stuff I want to do, I want to help people… (B25)

## Theme 4: Well-being

The above psychological processes were complemented by a more active engagement in managing well-being and lifestyle. A determination to stay well was identified.

…I swore from that day on, no man, money, love or beast would ever put me back into that situation again and I have stuck to it. (A15)

…I've been there, right there to the lowest of the low. And pulled myself back from it. And that's difficult to do. (C12)

This led to a greater engagement in well-being-related activities, including arts, music, sport, nature and learning.

I think the art has given me great kind of, great kind of structure. (A01)

…I've just found learning to be so therapeutic and rewarding that I feel like I am at a point now where I can actually study and put my mind towards…doing something worthwhile. (B10)

## Theme 5: Relationships

Many participants identified changes in relationships. For some this involved more actively choosing which relationships to continue, to re-start or to end.

I got rid of this awful man in my life. (B06)

I needed to go back to a couple of my old primary school friends' houses and ask for their forgiveness for something. (B07)

For others, the change was in the value placed on relationships with others.

…it's brought the incredible closeness with people with mental health with my, with my immediate family and friends… (A03)

It's been a process of learning that I needed, I need desperately, I desperately needed family, you know, people I feel safe with, to be myself. (B09)

A greater ability to empathise with others was also identified.

So my purpose really is young people and even when I see the destruction that young people are going through I never blame them, I said it stems from somewhere… (A17)

You have understanding, empathy…You really empathise. (B15)

All of these relationship processes informed a desire to give back, both by supporting others in similar situations and by giving back to society. This code differs from the survivor mission and mutuality sub-themes in its emphasis on giving as intrinsically beneficial.

I just see myself as hopefully being a beacon to others who are, you know, struggling, and others who are finding things difficult. (B02)

I want to go on and help people if I can that have been through the same thing. (C18)

## Theme 6: Spirituality

Participants, especially in group A, described a deeper engagement with spirituality, religious and existential endeavours. For some, this was expressed as an enhanced spiritual awareness.

Now I'm sort of growing older I know they're spirit animals, I still have them and so it's the wolves that are the most powerful so I do feel protected by those. (A18)

…as I went through that whole process it was like this massive opening, just kind of spiritual opening again. (A10)

This was associated with new or renewed observable engagement in spiritual or religious practices.

… I was sort of meditating and looking at the more spiritual aspects of my life and…I was looking and seeking that help… (A08)

…just pray, pray. (laughs) It works, do it, don't be scared. (C19)

## DISCUSSION

Post-traumatic growth concepts were identified in 64 (83%) of the interviews with 77 diverse participants describing their mental health recovery, and no participants rejected the idea of post-traumatic growth. The six superordinate categories are not ordered, but one narrative consistent with the results is that post-traumatic growth involves learning about oneself (self-discovery) leading to a new sense of who one is (sense of self) and one's appreciation of life (life perspective). Observable positively-valued changes are a greater focus on self-management (Well-being) and more importance being attached to relationships (relationships) and spiritual/religious engagement (spirituality). Individuals may start from any point in this narrative, so an alternative description would be that individuals experience a change in their life perspective, enabling an alternative Sense of self to be developed, which facilitates new kinds of self-discovery. The conceptual framework is compatible with the five growth processes identified in event trauma research, but participants also experienced changes in

(a) self-discovery, (b) sense of self, specifically including integration of illness-related experiences into identity and (c) the importance they attached to active self-management of well-being. Many participants were currently struggling with adversity so the term 'post-traumatic' may need amending, perhaps to 'trauma-related' for those who view their experiences as related to trauma.

The strengths of the study include the relatively large sample size for a qualitative study, the purposive sampling of diverse and under-researched participant groups, and the large analyst team (n=8) bringing multiple perspectives to enhance trustworthiness of data analysis. Weaknesses include the limited geographical spread mainly from two parts of England, the absence of member-checking, not collecting data on historical mental health service use by group A participants, and an assumption that trauma was present and therefore not asked about in interview (allowing assessment of the extent to which participants framed their experiences as 'trauma'). Identifying a change as positive was not always straightforward, so change experiences which in this study were not coded as positive psychological change may merit further investigation. These were not dissenting voices or competing accounts, but statements about types of change which were not judged by analysts to be post-traumatic growth. Examples which were not coded as post traumatic growth include alternative methods of getting well, escapism, knowing where to access support, cultural differences and environmental factors. Recovering from 'trauma' was not the language or frame of reference used by many participants in telling their stories, so this represents a researcher framing of experiences. However, many of the changes identified by participants were clearly describing growth arising directly out of their experiences. Relatedly, the extent to which participants attributed changes to post-traumatic growth, and the extent to which their formulations were derived from cultural understandings, were not explored. Our data are consistent with a view that the development of trauma-informed mental health services, with consequent changes in language and constructs used to describe experiences, will lead to an increase in individuals using post-traumatic growth concepts in their own recovery narratives. An alternative analysis approach would have involved wholly inductive coding followed by comparison with existing research. This would be more applicable to an interview study using a topic guide to assess previous experiences of trauma and identifying specifically post-traumatic growth experiences, rather than the current topic guide which was focused on recovery narratives and did not specifically ask about previous trauma experiences. The wide diagnostic spread of participants and high proportion (36%) who preferred not to give a diagnosis could be seen as a limitation. The standard health research approach would be to use a standardised assessment to confirm research diagnosis, but recovery research tends to have a more transdiagnostic focus. Finally, the cross-sectional coding inevitably abstracts

from complex and contextualised stories, so may not capture the ambivalent and conflicting perspectives common in narratives.

This study provides further evidence that post-traumatic growth concepts may be relevant to many people living with psychosis and other severe mental health problems, including previously under-research groups. A mental health service orientation towards supporting recovery is recommended internationally,[46 47] and central to national policy in many countries.[48–51] A recovery orientation involves system transformation[52] requiring new clinical approaches including a greater emphasis on supporting strengths,[53–55] self-management,[56 57] hope[58 59] and well-being,[60–62] more use of new interventions such as positive psychology,[63 64] Recovery Colleges[65–68] and peer support,[69 70] and a greater focus on human rights.[71 72] The current study supports the case that trauma-informed approaches to mental healthcare should be added to this list of recovery-supporting innovations.[73]

The conceptual framework provides a theoretical foundation for work to support post-traumatic growth in mental health. Encouraging post-traumatic growth is an important contribution to supporting recovery. For example, a study of post-traumatic growth and recovery in 34 people experiencing first episode psychosis concluded that 'people with early psychosis may benefit from disclosing their experiences of psychosis, including those aspects that were traumatic, as this may support the processes of recovery and post-traumatic growth'. (Pietruch and Jobson, p213)[74] A clinical implication is that it is important to reinforce efforts by service users to find their own personally satisfactory meaning of their experiences, rather than simply encouraging the adoption of a clinical explanatory model. Similarly, supporting service users to engage in well-being self-management, to actively choose relationships and to explore spiritual development should be a significant clinical focus.

The relationship between mental health recovery and post-traumatic growth is unclear, as is the extent to which they are the same or overlapping but distinct phenomena. Prospective longitudinal research is needed to investigate if there is a temporal relationship between positive psychological change and development of an identity as a 'person in recovery' for different clinical sub-populations, and how these change in the context of crisis or relapse. The implications for mental healthcare processes also need to be investigated, specifically identifying clinical sub-populations for whom a post-traumatic growth approach is particularly indicated, and developing and evaluating manualised treatment approaches which support the identified positive psychological changes. What is clear is that moving forward from severe mental health problems is not simply a question of taking treatment as prescribed, but a far more active process of engaging and re-engaging in the search for meaning for oneself, learning to manage the demands of living well, and finding one's place in the world.

## PATIENT AND PUBLIC INVOLVEMENT STATEMENT

Patients were involved in acquisition of funding, both through a lived experience advisory panel (LEAP) informing the design and through involvement as applicants. Several interviewers and analysts had their own lived experience of mental ill-health and recovery, in addition to their professional training. The NEON LEAP comprising 10 members with lived experience of mental ill-health and recovery informed the ethics application, trained the interviewers, informed the topic guide and supported access to group D. A LEAP member was involved in interpretation of the findings and as a co-author of this paper. The LEAP are leading the writing of a guide to sharing stories which will be informed by this research.

**Author affiliations**
[1]Institute of Mental Health, University of Nottingham School of Health Sciences, Nottingham, UK
[2]Department of Psychology, University of Nottingham, Nottingham, UK
[3]Institute of Mental Health, NEON Lived Experience Advisory Panel, Nottingham, UK
[4]Health Service and Population Research Department, Institute of Psychiatry, Psychology and Neuroscience, King's College London, London, UK
[5]Unit of Social and Community Psychiatry, Queen Mary University of London, London, UK
[6]School of Health Sciences, University of Nottingham, Nottingham, UK
[7]Department of Community Mental Health, University of Haifa, Haifa, Israel

**Acknowledgements** MS acknowledges the support of Center for Mental Health and Substance Abuse, University of South-Eastern Norway and the NIHR Nottingham Biomedical Research Centre (BRC). GT is supported by NIHR Collaboration for Leadership in Applied Health Research and Care South London (King's College London NHS Foundation Trust), Department of Health via the NIHR BRC and Dementia Unit (South London and Maudsley NHS Foundation Trust in partnership with King's College London and King's College Hospital NHS Foundation Trust), European Union Seventh Framework Programme (FP7/2007-2013, Emerald project), National Institute of Mental Health (National Institutes of Health R01MH100470, Cobalt study) and UK Medical Research Council (MR/S001255/1 Emilia; MR/R023697/1 Indigo Partnership) awards.

**Contributors** MS and LB made a substantial contribution to the conception or design of the work. MS, SR-E, LB, JL-B, DF, AH, GT, RM, KP, SP, AR, DR and ED contributed to the acquisition, analysis or interpretation of data for the work. MS, SR-E, LB, JL-B, DF, AH, GT, RM, KP, SP, AR, DR and ED were involved in drafting the work or revising it critically for important intellectual content, and gave final approval of the version to be published. MS, SR-E, LB, JL-B, DF, AH, GT, RM, KP, SP, AR, DR and ED agree to be jointly accountable for all aspects of the work.

**Funding** This article is independent research funded by the National Institute for Health Research (NIHR) under its Programme Grants for Applied Research Programme (Narrative Experiences ONline (NEON) Programme, RP-PG-0615-20016).

**Disclaimer** The views expressed are those of the authors and not necessarily those of the NHS, the NIHR, the Department of Health and Social Care or other supporting funders.

**Competing interests** None declared.

**Patient consent for publication** Not required.

**Provenance and peer review** Not commissioned; externally peer reviewed.

**Data sharing statement** No additional data are available.

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
