## [Reviewer comments · BMJ Open]

ARTICLE DETAILS

TITLE (PROVISIONAL)	Post-traumatic growth in mental health recovery: qualitative study of narratives
AUTHORS	Slade, Mike; Rennick-Egglestone, Stefan; Blackie, Laura; Llewellyn-Beardsley, Joy; Franklin, Donna; Hui, Ada; Thornicroft, Graham; McGranahan, Rose; Pollock, Kristian; Priebe, Stefan; Ramsay, Amy; Roe, David; Deakin, Emilia

VERSION 1 - REVIEW

REVIEWER	Carol Harvey Professor of Psychiatry & Consultant Psychiatrist Department of Psychiatry, University of Melbourne and NorthWestern Mental Health, Victoria, Australia
REVIEW RETURNED	14-Feb-2019

GENERAL COMMENTS	This is a well-written manuscript on a topic of interest. Some of the strengths of this study include: the purposive sampling of under-researched participants; and the incorporation of lived experience perspectives within the research team. I suggest that the major issue which should be addressed by the authors is outlined in my major comments concerning Results and Discussion below. In brief, the issues are (as flagged by the authors), the use of a preliminary coding framework rather than an inductive approach to analysis, and that recovery from trauma was primarily a researcher framing of experiences, together with the interpretations and conclusions linked with this (i.e. the trustworthiness of data analysis and interpretation). The current interpretation of these findings is that they provide evidence that post-traumatic growth concepts are relevant to many people living with psychosis and other severe mental health problems. However, is it not the case that an equally plausible interpretation is that the findings add to our understandings of recovery processes more generally, including those relevant to under-researched groups, rather than specifically to post-traumatic growth processes in mental health? Minor comments Title: Page 1, line 3, the second half of the title could be more descriptive than 'qualitative study', for example referring to narrative approach. Abstract: Page 2, line 24, design should include brief analytic details. Page 2, line 56, results describe the life perspective category as "one's relationship with life"; however, consistent with the description in Table 2, suggest this could read "one's appreciation of life".
--

	Page 3, lines 10-24, conclusions – suggest these are reviewed in light of my major comments on the Discussion within the article. Introduction: Page 7, line 45, introduction – “...interviews all identify...” should read “...interviews all identified...” Methods: Page 9, line 45, Methods, Group C – can the authors specify how an area with less than 10,000 population was defined? Was this a local government area, electoral district or similar? Page 10, lines 56-59 and page 11, lines 3-5, Procedures – suggest reverse the order of these two sentences and modify them accordingly so as to place the participants first. Page 10, Methods, Procedures – suggest the research paradigm is explicitly articulated here. Results: Page 18, lines 40-47, Results – is this a sub-category of the Relationships category? Discussion: Page 20, line 12, Discussion – same comment about whether to describe the life perspective category as “one’s relationship with life” or as “one’s appreciation of life”. Page 37, Table 2, line 13 – suggest main definition of sense of self should be in bold, consistent with the other main definitions in the Table. Page 37, Table 2, line 22 – “Increased determined...” needs review and revision Major comments: Introduction: Page 7, Introduction, can the preliminary early psychosis-specific processes which may represent post-traumatic growth be more clearly described? It was difficult to discern the link between this literature and the choice of corresponding terms in the preliminary coding framework. Methods: Page 10, line 43, Procedures and online supplement 1, see previous comment about whether a clearer link can be drawn between this statement, the preliminary coding framework and the literature for greater transparency. Pages 11 & 12, Analysis, were participants involved in checking the data or reviewing the analysis? If not, this should be reported as a limitation. Results: Were there any dissenting voices/competing accounts in this study? These were not evident, and it seems plausible that some participants may not have been able to identify and describe growth, whether or not linked with trauma experiences. The method reports that field notes were taken and were included in the analysis; however, the description of results does not make it clear how field notes contributed to the findings. Discussion: Page 21, lines 4-6, Discussion, refer to “change experiences which in this study were not coded as positive psychological change may merit further investigation”. Linked with my previous comment, are these some of the dissenting voices/competing accounts? If yes, these either need to be described in the Results or if this is not possible, this needs more explicit description as a limitation of this study. Page 21, lines 12-17, Discussion, describes recovering from trauma not being the frame of reference used by many participants and representing primarily a researcher framing of experiences.
--	--

	This is somewhat problematic and may link with my previous comments about the lack of competing accounts. Could the authors address this please? Further, as the authors go on to describe, an alternative analysis approach could have been adopted and would have been preferable in terms of allowing participants' voices to come through more strongly. I suggest the authors either need to: (1) justify their choice of method more and be more explicit about the limitations of their approach and revise their discussion accordingly (for example, page 22, lines 3-15); or (2) re-analyse using the alternative approach.
--	---

REVIEWER	Dr Breda Cullen University of Glasgow, UK
REVIEW RETURNED	09-Mar-2019

GENERAL COMMENTS	Thank you for the opportunity to read this manuscript. The aim of this study was to develop a conceptual framework for PTG in the context of recovery, in adults who have experienced psychosis and other severe mental health problems. The participants comprised four distinct samples, with different kinds of lived experience of mental health problems, service use, and peer support roles. Most participants showed evidence of PTG in their recovery narratives. The authors identified six superordinate thematic categories: Self-discovery, Sense of self, Life perspective, Wellbeing, Relationships, and Spirituality. The authors conclude that PTG in the context of mental health recovery, as distinct from PTG following other kinds of trauma, involves a greater emphasis on self-discovery, illness-related experiences and self-management of wellbeing. On the whole, this is a well written and engaging paper which addresses an important topic. My specific suggestions for amendments are as follows: Introduction The authors do not refer here to their own previous review and conceptual framework of personal recovery in mental health (Leamy et al, 2011). This is cited later in the paper, but I think it should be introduced alongside the other literature earlier on. Although this framework is about recovery rather than growth specifically, there is important overlap in some of the concepts, and it also appears that the authors may have drawn on this when constructing their initial coding framework for this study. There is a typo in the meta-analysis CI on p6 (OR2.78, 95%CI 5 2.34–3.31). Methods For Group A, what do the authors mean by “self-identified lifetime experience of psychosis”? Is this anyone who has ever had a psychotic-type symptom, regardless of duration or frequency? It would be very helpful to know more about how this study was presented to potential participants, as this will have influenced
---

decisions to take part. For example, was it presented as a study of recovery, or growth, or experiences in general? Can the authors provide copies of the adverts used for all four groups, as supplementary materials?

Results

It would be informative to know how many in Group A had accessed secondary MH services in the past, prior to the five-year period in the eligibility criteria.

The authors noted in the introduction that abuse and other kinds of serious adversity are very common in people with mental ill-health. It is unfortunate that no information is presented about whether the participants in this study had ever experienced other specific adverse circumstances or 'event trauma', and, if so, whether this was incorporated into their recovery story.

It isn't clear whether Groups B, C and D were required to have had experience of psychosis or other severe MH problems, which is the explicit focus of this paper. Table 1 indicates a range of primary diagnoses, some of which do not necessarily fit under the rubric of psychosis or other severe MH problems (e.g. anxiety, dysthymia, ADHD, autism). More than one-third preferred not to give information about a primary diagnosis. This presents some limitations with regard to the stated focus of this paper, and the ability of the reader to evaluate the applicability of the findings more generally. In light of this, can the authors provide a clearer justification for the decision to call their resulting framework "a conceptual framework for post-traumatic growth in psychosis and other severe mental health problems"?

Discussion

Related to my comment above regarding adversity/trauma, I note that the authors state on p20 that there was "an assumption that trauma was present and therefore not coded for by analysts". This is acknowledged as a weakness by the authors, but I think this warrants a little more discussion, because it relates to an important underlying issue of what 'trauma' is, and whether this is interchangeable with 'challenging life circumstances'. Why did the analysts assume trauma was present for everyone, just because they had experienced mental ill-health? Would the participants have agreed with this assumption? Similarly, the authors suggested rephrasing 'post-traumatic' as 'trauma-related', but this again assumes an implicit trauma narrative for everyone with mental ill-health. I appreciate that the authors have acknowledged this issue to some extent on pp20-21, but I would like to see this expanded a little more.

I am not clear why the authors think that paired family or mental health staff interviews would be particularly important in future research; is there evidence from PTG research in other populations that outside perspectives provide important new insights?

On p21, can the authors clarify what they mean by "the extent to which their formulations were derived from cultural tropes"?

VERSION 1 – AUTHOR RESPONSE

Reviewer 1

- I suggest that the major issue which should be addressed by the authors is outlined in my major comments concerning Results and Discussion below. In brief, the issues are (as flagged by the authors), the use of a preliminary coding framework rather than an inductive approach to analysis, and that recovery from trauma was primarily a researcher framing of experiences, together with the interpretations and conclusions linked with this (i.e. the trustworthiness of data analysis and interpretation).

We address this reviewer's very insightful questions in responses below.

- The current interpretation of these findings is that they provide evidence that post-traumatic growth concepts are relevant to many people living with psychosis and other severe mental health problems. However, is it not the case that an equally plausible interpretation is that the findings add to our understandings of recovery processes more generally, including those relevant to under-researched groups, rather than specifically to post-traumatic growth processes in mental health?

We have softened our interpretation and now note the relevance particularly to under-researched groups (Discussion, 'This study provides...' sentence 1).

- Page 1, line 3, the second half of the title could be more descriptive than 'qualitative study', for example referring to narrative approach.

Amended as suggested (Title).

- Page 2, line 24, design should include brief analytic details.

Amended as suggested (Abstract Design).

- Page 2, line 56, results describe the life perspective category as "one's relationship with life"; however, consistent with the description in Table 2, suggest this could read "one's appreciation of life".

Amended as suggested (Abstract Results).

- Page 3, lines 10-24, conclusions – suggest these are reviewed in light of my major comments on the Discussion within the article.

Amended as suggested (Abstract Conclusions).

- Page 7, line 45, introduction – "...interviews all identify..." should read "...interviews all identified..."

Amended as suggested (Introduction, 'There is only...' last sentence).

- Page 9, line 45, Methods, Group C – can the authors specify how an area with less than 10,000 population was defined? Was this a local government area, electoral district or similar?

Amended as suggested (Methods Participants, 'Group C' last sentence).

- Page 10, lines 56-59 and page 11, lines 3-5, Procedures – suggest reverse the order of these two sentences and modify them accordingly so as to place the participants first.

Amended as suggested (Methods Procedures, 'Each participant...').

- Page 10, Methods, Procedures – suggest the research paradigm is explicitly articulated here.

Added as suggested (Methods Analysis, 'The four coders...' sentence 3ff)

- Page 18, lines 40-47, Results – is this a sub-category of the Relationships category?

Clarified as suggested (Results, 'Theme 5: Relationships').

- Page 20, line 12, Discussion – same comment about whether to describe the life perspective category as "one's relationship with life" or as "one's appreciation of life".

Amended as suggested (Discussion, 'Post-traumatic growth concepts...' sentence 2).

- Page 37, Table 2, line 13 – suggest main definition of sense of self should be in bold, consistent with the other main definitions in the Table.

Amended as suggested (Table 2).

- Page 37, Table 2, line 22 – "Increased determined..." needs review and revision

Now corrected (Table 2).

- Page 7, Introduction, can the preliminary early psychosis-specific processes which may represent post-traumatic growth be more clearly described?

Elaborated as suggested (Methods, 'There is strong evidence...', sentence 4ff).

- It was difficult to discern the link between this literature and the choice of corresponding terms in the preliminary coding framework.

The preliminary coding framework development process has been elaborated as suggested (Methods Procedures, 'A preliminary coding framework...' sentences 1-2).

- Page 10, line 43, Procedures and online supplement 1, see previous comment about whether a clearer link can be drawn between this statement, the preliminary coding framework and the literature for greater transparency.

The preliminary coding framework development process has been elaborated as suggested (Methods Procedures, 'A preliminary coding framework...' sentences 1-2).

- Pages 11 & 12, Analysis, were participants involved in checking the data or reviewing the analysis? If not, this should be reported as a limitation.

Now reported as a limitation, as suggested (Discussion, 'The strengths...' sentence 2).

- Were there any dissenting voices/competing accounts in this study? These were not evident, and it seems plausible that some participants may not have been able to identify and describe growth, whether or not linked with trauma experiences.

Clarified as suggested (Discussion, 'Post-traumatic growth concepts...' sentence 1; Discussion, 'The strengths...' sentence 4).

- The method reports that field notes were taken and were included in the analysis; however, the description of results does not make it clear how field notes contributed to the findings.

Clarified as suggested (Methods Analysis, 'Coding was initially...' sentence 1).

- Page 21, lines 4-6, Discussion, refer to "change experiences which in this study were not coded as positive psychological change may merit further investigation". Linked with my previous comment, are these some of the dissenting voices/competing accounts? If yes, these either need to be described in the Results or if this is not possible, this needs more explicit description as a limitation of this study.

It is now clarified what these mean (Discussion, 'The strengths...' sentence 4).

- Page 21, lines 12-17, Discussion, describes recovering from trauma not being the frame of reference used by many participants and representing primarily a researcher framing of experiences. This is somewhat problematic and may link with my previous comments about the lack of competing accounts. Could the authors address this please?

We thank the reviewer for highlighting the incompleteness of our description. We now clarify that the main point we make is that participants were clearly describing PTG concepts in their narratives (Discussion, 'The strengths...' sentence 7 'However,...') but not using the language of PTG in their narratives (Discussion, 'The strengths...' sentence 6 'Recovering from...'), so we suggest that changes in language and constructs which would arise from a growth in trauma-informed services

would likely increase the use of PTG concepts in recovery narratives (Discussion, 'The strengths...' sentence 9 'Our data...').

- Further, as the authors go on to describe, an alternative analysis approach could have been adopted and would have been preferable in terms of allowing participants' voices to come through more strongly. I suggest the authors either need to: (1) justify their choice of method more and be more explicit about the limitations of their approach and revise their discussion accordingly (for example, page 22, lines 3-15); or (2) re-analyse using the alternative approach.

We now clarify that an inductive approach would be more applicable if the topic guide had been more focussed on post-traumatic growth concepts (Discussion, 'The strengths...' sentence 11 'This would be...').

Reviewer 2

- The authors do not refer here to their own previous review and conceptual framework of personal recovery in mental health (Leamy et al, 2011). This is cited later in the paper, but I think it should be introduced alongside the other literature earlier on. Although this framework is about recovery rather than growth specifically, there is important overlap in some of the concepts, and it also appears that the authors may have drawn on this when constructing their initial coding framework for this study.

Amended as suggested (Introduction, 'The concept of recovery...').

- There is a typo in the meta-analysis CI on p6 (OR2.78, 95%CI 5 2.34–3.31).

Now corrected (Introduction, 'Post-traumatic growth has relevance...' sentence 5).

- For Group A, what do the authors mean by "self-identified lifetime experience of psychosis"? Is this anyone who has ever had a psychotic-type symptom, regardless of duration or frequency?

Now clarified (Methods Participants, 'Group A...' sentence 1).

- It would be very helpful to know more about how this study was presented to potential participants, as this will have influenced decisions to take part. For example, was it presented as a study of recovery, or growth, or experiences in general? Can the authors provide copies of the adverts used for all four groups, as supplementary materials?

A range of adverts were used for each group depending on the advertising route, so we now provide the advertising guidelines (Online Supplement 1).

- It would be informative to know how many in Group A had accessed secondary MH services in the past, prior to the five-year period in the eligibility criteria.

We did not collect this data and now note this as a limitation (Discussion, 'The strengths...' sentence 2).

- The authors noted in the introduction that abuse and other kinds of serious adversity are very common in people with mental ill-health. It is unfortunate that no information is presented about whether the participants in this study had ever experienced other specific adverse circumstances or 'event trauma', and, if so, whether this was incorporated into their recovery story.

We did not collect this data and now note this as a limitation (Discussion, 'The strengths...' sentence 2) and identify how future research should address this important question (Discussion, 'The strengths...' sentence 11 'This would be more applicable...').

- It isn't clear whether Groups B, C and D were required to have had experience of psychosis or other severe MH problems, which is the explicit focus of this paper. Table 1 indicates a range of primary diagnoses, some of which do not necessarily fit under the rubric of psychosis or other severe MH problems (e.g. anxiety, dysthymia, ADHD, autism). More than one-third preferred not to give information about a primary diagnosis. This presents some limitations with regard to the stated focus of this paper, and the ability of the reader to evaluate the applicability of the findings more generally. In light of this, can the authors provide a clearer justification for the decision to call their resulting framework "a conceptual framework for post-traumatic growth in psychosis and other severe mental health problems"?

Groups B and C were using (successfully or not) secondary mental health services, which in the UK aligns with having severe mental health problems. Group D involved people whose mental health problems had been a sufficiently important part of their history that they were using their experiences as the foundation for their peer role, again indicating that their mental health problems were severe. We now comment on the diagnostic spread (Discussion, 'The strengths...' sentences 12-13 'The wide diagnostic spread...'). We suggest not to comment explicitly on the 'severe' issue, since the concept of 'severe' is widely used, contested when examined in detail, and beyond the scope of this study.

- Related to my comment above regarding adversity/trauma, I note that the authors state on p20 that there was "an assumption that trauma was present and therefore not coded for by analysts". This is acknowledged as a weakness by the authors, but I think this warrants a little more discussion, because it relates to an important underlying issue of what 'trauma' is, and whether this is interchangeable with 'challenging life circumstances'. Why did the analysts assume trauma was present for everyone, just because they had experienced mental ill-health? Would the participants have agreed with this assumption?

We now note the limitation that we did not investigate participant agreement with this assumption (Discussion, 'The strengths...' sentence 2), and identify how future research might better address this important question (Discussion, 'The strengths...' sentence 11 'This would be more applicable...').

- Similarly, the authors suggested rephrasing 'post-traumatic' as 'trauma-related', but this again assumes an implicit trauma narrative for everyone with mental ill-health. I appreciate that the authors have acknowledged this issue to some extent on pp20-21, but I would like to see this expanded a little more.

We now clarify that 'trauma-related' is useful for people with an implicit trauma narrative (Abstract Conclusions, sentence 3; Discussion, 'Post-traumatic growth concepts...' final sentence).

- I am not clear why the authors think that paired family or mental health staff interviews would be particularly important in future research; is there evidence from PTG research in other populations that outside perspectives provide important new insights?

Given the addition and amplification of many limitations, we have deleted this limitation.

- On p21, can the authors clarify what they mean by "the extent to which their formulations were derived from cultural tropes"?

We have replaced the term 'trope' and clarified our meaning (Discussion, 'The strengths...' sentence 8-9 'Relatedly, the extent...').

VERSION 2 – REVIEW

REVIEWER	Carol Harvey Professor of Psychiatry & Consultant Psychiatrist Department of Psychiatry, University of Melbourne & NorthWestern Mental Health Melbourne Victoria Australia
REVIEW RETURNED	07-May-2019

GENERAL COMMENTS	Thank you for clarifying and addressing the issues raised. I note one typo in the revision, as follows: (Discussion, third paragraph "This study provides.....". In this first sentence, under-research should read under-researched.
---

REVIEWER	Breda Cullen University of Glasgow, UK
REVIEW RETURNED	16-Apr-2019

GENERAL COMMENTS	Thank you for addressing my comments. I believe the revisions are satisfactory and the paper should be accepted.
--